# Cumulative route improvements spontaneously emerge in artificial navigators even in the absence of sophisticated communication or thought

**Edwin S. Dalmaijer**● *

School of Psychological Science, University of Bristol, Bristol, United Kingdom

* edwin.dalmaijer@bristol.ac.uk

## Abstract

Homing pigeons (*Columba livia*) navigate by solar and magnetic compass, and fly home in idiosyncratic but stable routes when repeatedly released from the same location. However, when experienced pigeons fly alongside naive counterparts, their path is altered. Over several generations of turnover (pairs in which the most experienced individual is replaced with a naive one), pigeons show cumulative improvements in efficiency. Here, I show that such cumulative route improvements can occur in a much simpler system by using agent-based simulation. Artificial agents are in silico entities that navigate with a minimal cognitive architecture of goal-direction (they know roughly where the goal is), social proximity (they seek proximity to others and align headings), route memory (they recall landmarks with increasing precision), and continuity (they avoid erratic turns). Agents' behaviour qualitatively matched that of pigeons, and quantitatively fitted to pigeon data. My results indicate that naive agents benefitted from being paired with experienced agents by following their previously established route. Importantly, experienced agents also benefitted from being paired with naive agents due to regression to the goal: naive agents were more likely to err towards the goal from the perspective of experienced agents' memorised paths. This subtly biased pairs in the goal direction, resulting in intergenerational improvements of route efficiency. No cumulative improvements were evident in control studies in which agents' goal-direction, social proximity, or memory were lesioned. These 3 factors are thus necessary and sufficient for cumulative route improvements to emerge, even in the absence of sophisticated communication or thought.

## Introduction

Cumulative cultural evolution occurs when individuals pass down adaptive innovations through social means (e.g., teaching or copying), leading to progressive increases in fitness over generations [1,2]. In humans, this "ratcheting" [3] of socially transmitted improvements is vital to human technological advancement [4] and has historically been attributed to

**Data Availability Statement:** All code and data has been made publicly available through open repositories on GitHub (https://github.com/ esdalmaijer/artificial_navigators) and Zenodo (data:

https://doi.org/10.5281/zenodo.6944185; code: https://doi.org/10.5281/zenodo.10997495).

**Funding:** The author(s) received no specific funding for this work.

**Competing interests:** The authors have declared that no competing interests exist.

uniquely human "high-fidelity" communication [5]. However, experimental work has shown simple emulative learning is sufficient for cumulative culture to occur [6,7]. Some argue that other species show cumulative culture, e.g., in transmission chains of songs in zebra finches [8] and humpback whales [9], tool use in crows [10] and chimpanzees [11–13], and pattern reproduction in great tits [14] and baboons [15]. One particularly striking example comes from pigeons, which seem to pass down route improvements [16].

Homing pigeons (*Columba livia*) are suboptimal navigators that develop and remember idiosyncratic routes when flying alone or in pairs [17]. While paired birds fly more efficient routes than individuals [18], pairs in which experienced pigeons are swapped for naive ones show "innovation": beneficial modifications between generations [16] that meet criteria for cumulative culture [19]. Perhaps pigeons pool information between individuals, learn and decide through collective intelligence, and evaluate performance to prune worse innovations [16]; or develop intra-pair dynamics of communication and leadership [20].

An alternative explanation is that cumulative route improvements emerged as accidental by-product during navigation in groups. Here, I directly address this question using a minimal cognitive architecture in artificial agents that are bound by only 4 rules derived from avian navigation (Fig 1). The first is goal direction, akin to birds' solar [21] and magnetic compasses [22] that allow them to orient towards their home even from unfamiliar release sites unless under total overcast with disorienting magnets glued to their head [23]. The second is social proximity, which birds seek when flying together [24]. The third is route memory, which in pigeons could depend on visual landmarks [25] and improves over consecutive flights [26]. The fourth is continuity, a tendency to continue along the current heading to avoid implausibly erratic patterns. Crucially, there is no communal decision-making, evaluation of outcomes, or deliberate social communication.

The artificial navigator model is a weighted mixture of Von Mises distributions $\Phi$, with weights $w$ that add up to 1 (Eq 1). These are akin to normal distributions, but they are circular, so that the tails wrap around. To produce the next heading $h$ in journey $i$ at time $t+1$, an agent combines information from time $t$ on bearings towards the goal $b_{goal}$, the next memorised landmark $b_{landmark}$, and other agents' estimated future position $\hat{b}_{other}$. As in birds, not all bearings are equally precise, which is reflected in each component's precision parameter $\kappa$. For example, there is uncertainty about where the (solar/magnetic compass) goal is [21,22], whereas pigeon visual acuity is good enough [27,28] to identify nearby visual landmarks along a well-memorised route (although they are not always used, [29]). To prevent unnaturally jerky movements, the final component ensures continuity by sampling from a narrow distribution that is centred on the current heading. For a full account of the algorithm, please refer to Materials and methods.

$$
\begin{aligned}
h(i, t + 1) = {} & w_{goal}\Phi(b_{goal}, \kappa_{goal}) \\
& + w_{social}\Phi(\hat{b}_{other}, \kappa_{social}) \\
& + w_{memory}\Phi(b_{landmark}, \kappa_{memory,i}) \\
& + w_{continuity}\Phi(h(i, t), \kappa_{continuity})
\end{aligned}
\tag{1}
$$

Agents travelled in 3 conditions that mapped onto work in pigeons [16]: solo, paired, and an experimental condition with generational turnover. In the solo and pair conditions, 1 or 2 agents made 60 consecutive journeys. The experimental condition also involved pairs, but a naive replaced an experienced agent every 12 journeys. A total of 50 repetitions were done for each condition and set of weight parameters. Precision parameters were fixed at $\kappa_{continuity} = 8.69$ (equivalent SD = 0.35), $\kappa_{goal} = 1.54$ (1.0), $\kappa_{social} = 2.18$ (0.80), $\kappa_{memory,1} = 0.85$ (0.9) to

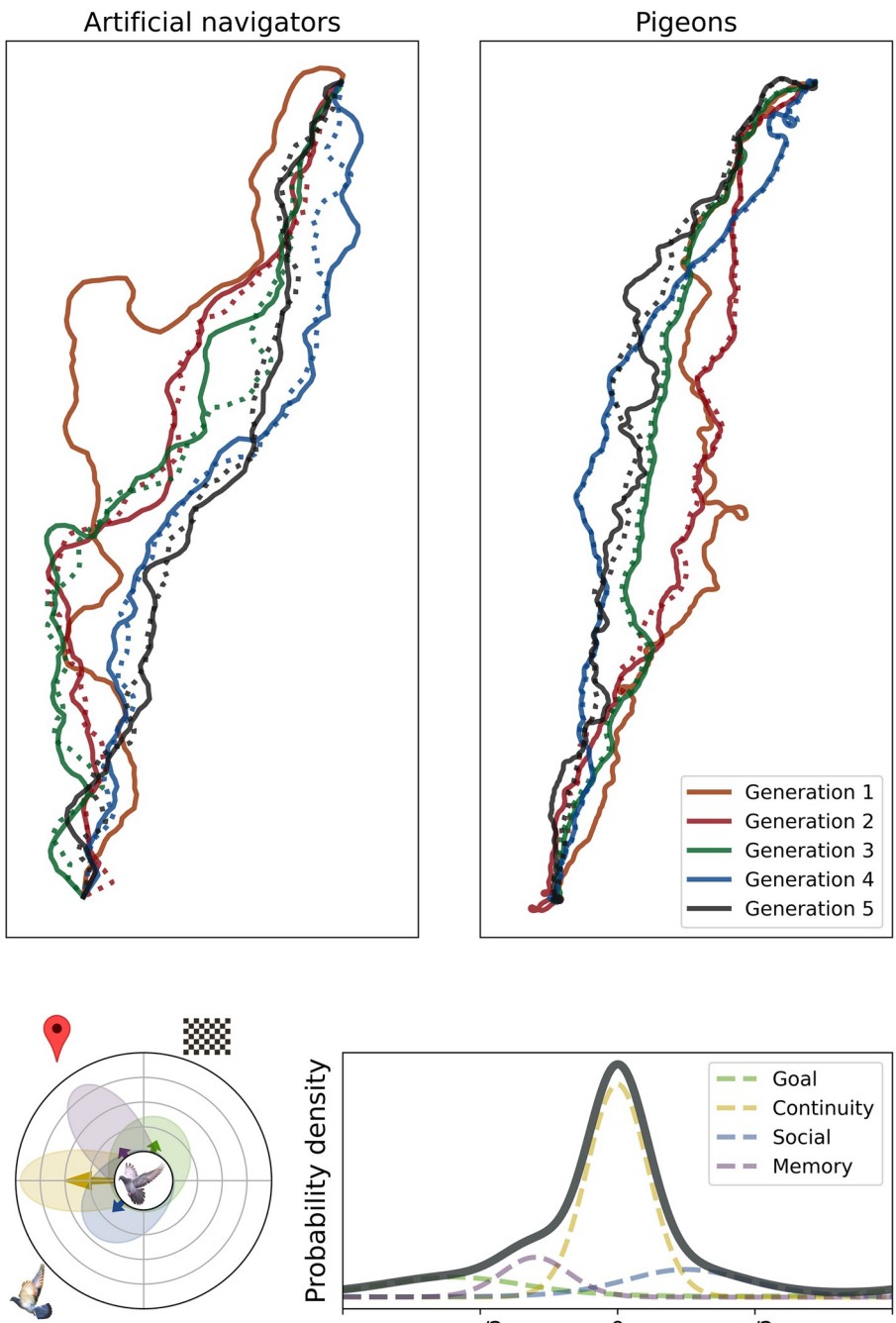

**Fig 1. The top panel shows paths from artificial agents (introduced here) and from pigeon data published by others** [20]**.** Each line represents the final flight in a generation. The first generation comprises a single individual; a naive individual was added in the second; and in all later generations the most experienced was replaced with a naive individual. Solid lines show lone or experienced individuals, dotted lines show naive ones. The bottom panel shows how agents navigated by sampling from a weighted mixture of Von Mises distributions. These were centred on bearings towards the goal (green), other agents (blue), landmarks along a memorised route (purple), and the previous heading (yellow). Bottom left shows these distributions in a radial plot, with arrows indicating component centres and weights. Bottom right shows the distributions and their weighted sum (black). Artwork used in this figure exists in the public domain or was released under a CC0 license and can be found on Wikimedia Commons (https://commons.wikimedia.org/) under file names F1_chequered_flag.svg, Google_Maps_pin.svg, RockDove.jpg, and Black_rock_pigeon.jpg.

$\kappa_{memory,5}$ = 6.78 (0.40), based on model fits for pigeon data. Note that memory precision improves over journeys, as per evidence from pigeon flight [26].

## What is (not) cumulative culture?

Human's accumulated innovations are undeniably "superlative" [30] and many depend on combining physical phenomena ("Type II" cultural evolution), whereas nonhuman innovations typically optimise only within a phenomenon ("Type I") [4]. An appealing framework describes 4 core criteria of cumulative cultural evolution: behaviour needs to **(1)** show variation introduced by interaction between individuals; **(2)** be passed on through social learning; **(3)** improve performance; and **4)** repeat over generations [19]. Few examples of "cumulative culture" in animals meet all 4, and rarely are extended criteria (functional dependence, diversification into lineages, recombination across lineages, exaptation, or niche construction) met [19].

Route improvements in successive generations of pigeons were described as cumulative culture [16], and it was indeed listed as meeting all core criteria of the aforementioned framework [19]. However, whether animals genuinely show cumulative culture is controversial. An alternative explanation is that individuals attend to others' actions, and then reinnovate a "latent solution" to produce similar outcomes [5]. Alternatively, apparent innovations could have previously been unobserved or learned not socially but in response to changing environments [31]. It is hotly debated whether these alternatives are valid and relevant; see [32] and the numerous responses for an overview of current opinions.

The agents employed in this study arguably do not meet the above standard. Their "innovation" is limited to an increase in efficiency, which is decidedly unlike the development of novel behaviour. While focussing on task efficiency offers insight into cumulative cultural evolution [33], a focus on task solutions can obscure that humans also actively discover new problems and generalise solutions between them, which nonhumans rarely do [34]. Agents also do not engage in "social learning" as it is traditionally defined: all they do is follow other individuals, without explicit demonstration or observation of a concrete task. Hence, I will refer to their outcome as "cumulative route improvements."

## Results

Artificial navigators travelled in an "experimental" condition with generational turnover (pairs with replacement of an experienced for a naive individual each generation) or in control conditions without turnover (paired or solo). They showed various levels of route efficiency (Figs 2 and A in S1 File), which was computed as start-goal distance divided by travelled distance [16], and ranged between 0 (never reached the goal) to 1 (straight line from start to goal). Parameters could be optimised for final-route efficiency (Fig 2, top) or improvement between generations (Fig 2, centre), and compared well to empirical pigeon data (Fig 2, bottom).

Cumulative route improvement was quantified as the increase in route efficiency between each generation. This occurred exclusively in the experimental condition (Figs A, B, and D in S1 File), replicating empirical data [16].

In the experimental condition, the average highest final-generation route efficiency was 0.884 (SD = 0.017, SEM = 2.47e-3) for parameters $w_{goal}$ = 0.25, $w_{social}$ = 0.20, $w_{memory}$ = 0.05, and $w_{continuity}$ = 0.50. Highest final efficiency (M = 0.875, SD = 0.021, SEM = 2.97e-3) was also achieved for these parameters in the pair condition, whereas final efficiency in the solo condition was highest at 0.863 (SD = 0.22, SEM = 3.10e-3) for parameters $w_{goal}$ = 0.35, $w_{social}$ = 0.05, $w_{memory}$ = 0.05, and $w_{continuity}$ = 0.55.

The highest average generational efficiency increase was 0.091 (SD = 0.014, SEM = 2.00e-3) and achieved at $w_{goal}$ = 0.15, $w_{social}$ = 0.25, and $w_{memory}$ = 0.40, $w_{continuity}$ = 0.20 in the

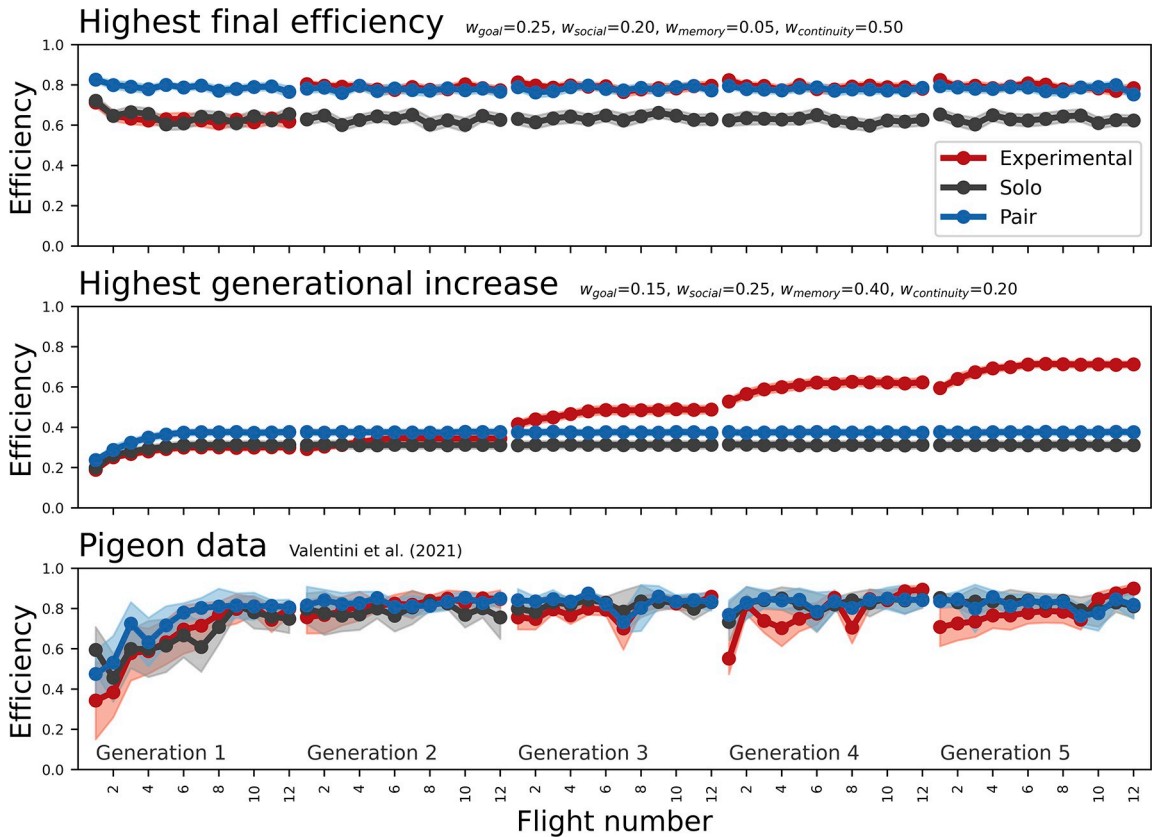

**Fig 2. Progression of route efficiency as a function of flight number.** The top panel shows results for the optimum for final efficiency, the middle for the optimum for intergenerational improvement, and the bottom panel for pigeon data published by others [20]. Lines show mean values over independent runs, with 95% confidence intervals as shaded areas. In the experimental condition, a naive agent replaced an experienced one in each generation; in the solo condition, a single agent made all journeys with no generational turnover; and in the pair condition, 2 agents journeyed together without turnover. Parameters for the navigation model were the same between each of the 3 conditions, and weights are listed above each panel.

experimental condition. For these parameters, intergenerational improvement was only 2.4e-5 (SD = 11.2e-3, SEM = 1.58e-4) in the pair condition, and 1.0e-4 (SD = 7.32e-4, SEM = 1.03e-4) in the solo condition. The highest generational turnover achieved in the pair condition was 0.013 (SD = 0.042, SEM = 5.90e-3 at $w_{goal}$ = 0.15, $w_{social}$ = 0.15, and $w_{memory}$ = 0.10, $w_{continuity}$ = 0.60) and 0.005 (SD = 0.019, SEM = 2.62e-3 at $w_{goal}$ = 0.15, $w_{social}$ = 0.20, and $w_{memory}$ = 0.10, $w_{continuity}$ = 0.55) in the solo condition. Out of 610 parameter combinations, 269 achieved a greater intergenerational improvement in the experimental condition than the highest pair control condition.

### Naive individuals can benefit from the experienced

In the experimental condition, naive individuals could benefit from following an experienced agent with established route memory. Compared to the pair control condition, naive individuals showed more efficient paths (Fig 3) if their experienced counterpart relied more strongly on memory ($w_{memory}$). However, naive agents were worse off at low memory-reliance, particularly if the relative influence of goal-direction ($w_{goal}$) was low, and if they more strongly sought social proximity ($w_{social}$).

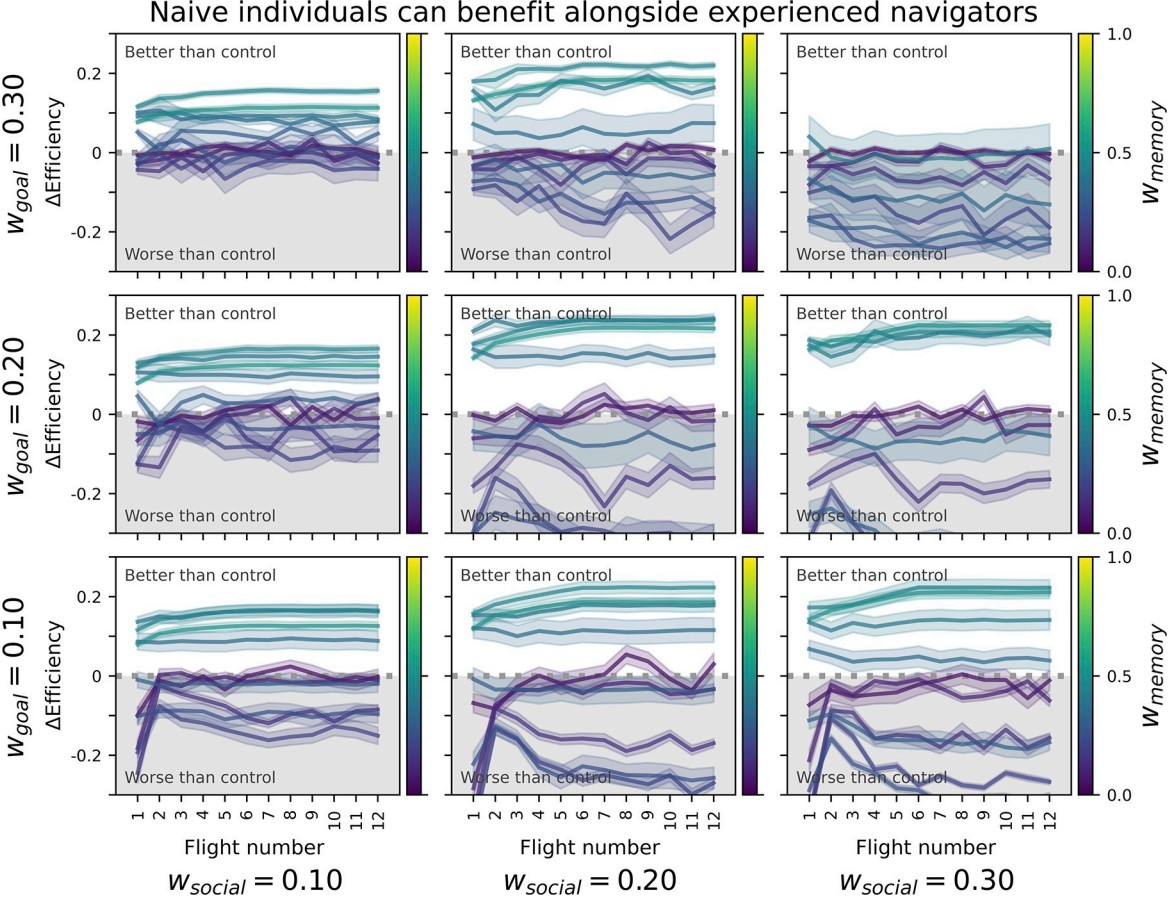

**Fig 3. Each panel shows the difference in route efficiency between naive agents in the experimental condition (generational turnover) and the first 12 journeys from agents in the pair control condition (without generational turnover).** Positive differences indicate that naive agents had better route efficiency compared to control. Each panel represents a combination of $w_{goal}$ and $w_{social}$ parameters, while darker lines indicate higher levels of $w_{memory}$. Lines represent averages across 50 independent runs and their shaded areas the 95% confidence interval.

## Experienced individuals benefit from the naive

While it is perhaps obvious that naive agents could benefit from following established paths, more surprising was that experienced individuals also benefitted from their naive counterparts. This occurred due to regression to the goal. Compared to extreme samples, random samples are more likely to be nearer a distribution's centre; this is regression to the mean. Similarly, experienced agents draw from internal distributions, including for goal-direction and route memory. Naive agents sample from internal distributions too, but do not have route memory yet, and hence are more biased towards the goal than experienced individuals. Because agents aim for social proximity, naive navigators should thus subtly pull experienced agents towards the goal.

This was born out empirically, as relative bearings for experienced towards naive agents were more likely to also be in the direction of the goal (Fig 4). This was primarily true for lower values of $w_{memory}$ and increased with $w_{social}$. Regression to the goal thus allowed naive agents to memorise slightly more efficient routes than their paired experienced agent.

## Control experiments with lesioned agents

Agents were lesioned in control experiments to investigate which navigation components were necessary for cumulative route improvements to emerge (Table 1). Control experiments

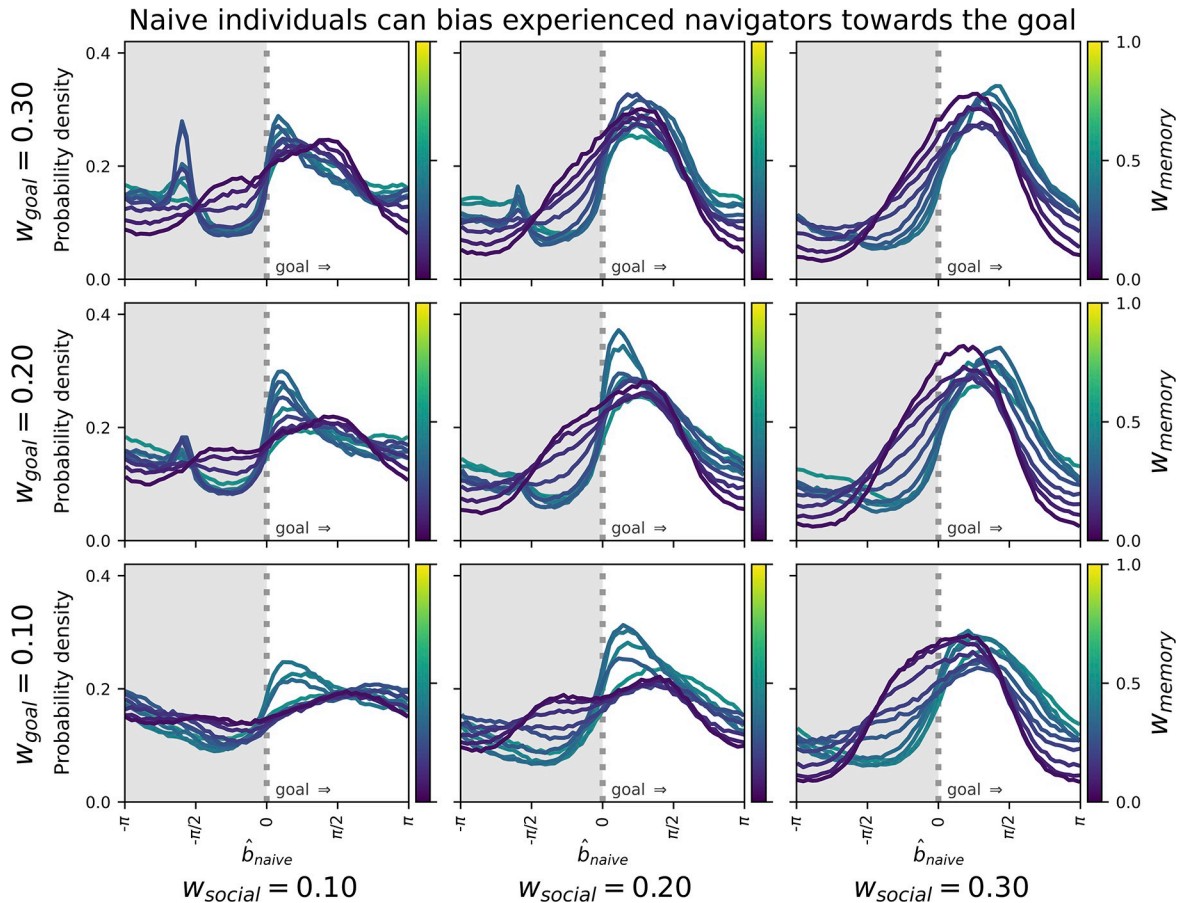

**Fig 4. Each panel shows the distribution of relative bearings towards the naive agent from the perspective of the experienced agent in generations 2–5 of the experimental condition.** Positive values on the x-axis indicate bearings towards the goal, and negative values bearings away from the goal. Distributions are generally right-heavy, indicating a bias of naive individuals to be positioned in the general direction of the goal. This tendency increases as a function of $w_{social}$ and to a lesser extent as a function of $w_{goal}$.

**Table 1. Efficiency quantifies how close agents were to the direct path from start to goal.** Final efficiency is measured in the last generation and generational increase as the difference between generations. Cumulative route improvements are reflected by a positive intergenerational increase and occur in the "experimental" condition ("pair" and "solo" are control conditions). The "no lesion" column reflects optimal scores; the other columns reflect scores after replacing the goal, social, memory, or continuity component with a uniform distribution (noise).

| | No lesion | Goal lesion (all) | Goal lesion (gen>1) | Social lesion | Memory lesion | Continuity lesion |
|---|---|---|---|---|---|---|
| **Final efficiency** | | | | | | |
| experimental | 0.884 | 0 | 0 | 0.794 | 0.894 | 0.178 |
| pair | 0.875 | 0 | 0.143 | 0.782 | 0.897 | 0.170 |
| solo | 0.790 | 0 | 0.074 | 0.788 | 0.837 | 0.173 |
| **Generational increase** | | | | | | |
| experimental | 0.091 | 0 | 7.95e-3 | 5.23e-4 | 8.99e-4 | 0.068 |
| pair | 2.37e-5 | 0 | −9.84e-4 | 1.96e-5 | 2.91e-3 | 2.96e-4 |
| solo | 1.05e-4 | 4.24e-6 | −2.84e-5 | −7.67e-5 | −8.04e-4 | 7.82e-5 |

**Table 2. This table illustrates how the precision of each navigation component impacts cumulative route improvements, which are quantified by a positive inter-generational increase in path efficiency in the "experimental" condition ("pair" and "solo" are control conditions).** The "normal precision" column reflects scores from the current model parameters. The "high precision" and "low precision" reflect halving and doubling the standard deviation, which is then transformed back into precision parameter κ for a navigational component Von Mises distribution.

| Generational increase | Very high precision | High precision | Normal precision | Low precision |
|---|---|---|---|---|
| **Goal** | $\kappa_{goal} = 101$ | $K_{goal} = 4.55$ | $K_{goal} = 1.54$ | $K_{goal} = 0.27$ |
| experimental | 0.073 | 0.085 | 0.091 | 0.058 |
| pair | 8.12e-4 | 1.05e-4 | 2.37e-5 | −5.79e-4 |
| solo | 7.01e-5 | 8.19e-6 | 1.05e-4 | −3.55e-5 |
| **Social** | $\kappa_{social} = 101$ | $K_{social} = 6.78$ | $K_{social} = 2.18$ | $K_{social} = 0.579$ |
| experimental | 0.092 | 0.093 | 0.091 | 0.032 |
| pair | −1.76e-4 | −3.05e-4 | 2.37e-5 | 3.02e-4 |
| solo | 1.69e-4 | −9.04e-5 | 1.05e-4 | 5.22e-4 |
| **Memory** | $\kappa_{memory} = 1e12$ | $K_{memory} = 25.5$ | $K_{memory} = 6.78$ | $K_{memory} = 2.18$ |
| experimental | 0.092 | 0.090 | 0.091 | 0.043 |
| pair | −2.63e-4 | −2.53e-4 | 2.37e-5 | 1.28e-3 |
| solo | −2.90e-4 | −5.69e-4 | 1.05e-4 | 2.56e-4 |
| **Continuity** | $\kappa_{continuity} = 401$ | $K_{continuity} = 33.2$ | $K_{continuity} = 8.69$ | $K_{continuity} = 2.67$ |
| experimental | 0.080 | 0.095 | 0.091 | 0.084 |
| pair | −2.54e-4 | −4.56e-4 | 2.37e-5 | 4.23e-4 |
| solo | 3.13e-4 | −1.39e-4 | 1.05e-4 | 2.01e-4 |

employed the same weights as those that achieved highest final efficiency or intergenerational efficiency increases in the experimental condition (described above). However, headings were sampled from uniform distributions (i.e., noise) instead of being influenced by goal-direction, social proximity, route memory, or continuity.

When goal-direction was lesioned for all generations, agents engaged in random walks that failed to reach the goal in time. When goal-direction was lesioned for all but the first generation, a path could be established within the first generation. This isolated goal-direction's necessity for intergenerational improvement, which should not occur with lesioned goal-direction if it is dependent on regression to the goal.

When social proximity or route memory was lesioned, efficiency was barely reduced, but generational increase was nullified. When continuity was lesioned, efficiency was greatly reduced, but the pattern of generational increases remained intact: present in the experimental condition, but not in pair or solo controls.

In another set of control experiments (Table 2), the precision of each navigational component was varied from low (wide Von Mises distribution) to very high (narrow distribution). Wider distributions reduced intergenerational efficiency increases, and a wide goal component even prevented agents from completing their routes. Narrower distributions effected less change, although a more precise goal component did subtly reduce increases in intergenerational efficiency. The pattern of cumulative route improvements in the experimental but not in the pair and solo control conditions was apparent throughout.

The lesion experiments suggest goal direction, social proximity, and route memory were all crucial for cumulative route improvements to emerge in this model.

## Artificial navigator model fits empirical data

When fitted on 10 repetitions of the experimental condition in pigeons (data published by [20]), average parameter estimates were $w_{goal} = 0.12$ (SEM = 0.03), $w_{social} = 0.16$ (SEM = 0.03), $w_{memory} = 0.09$ (SEM = 0.01), and $w_{continuity} = 0.59$ (SEM = 0.03). That these weights did not

align with optima for intergenerational improvement or efficiency for agents suggests that the artificial navigator model is insufficient to capture the full complexity of pigeon behaviour, which agrees with interpretations put forward by others [16,20].

## Discussion

The minimal cognitive architecture of goal-direction, social proximity, and long-term memory was sufficient for the emergence of cumulative route improvements. It was driven by regression to the goal over generations: as agents in a new pair aligned and converged their headings, experienced agents travelled along a remembered route, while their naive counterparts introduced a subtle goal-directed bias.

These results suggest that stepwise improvement between generations can occur when individuals simply seek proximity to others. Agents had no capacity or intent to communicate, but information transferred between them as naive agents followed and memorised experienced agents' routes, while their subtle goal-directed pull introduced stepwise improvement between generations. While previous work has demonstrated step-wise improvement between generations through emulative learning [6,7], tasks required strategic social learning and advanced cognitive skills. Another difference is that the current task presents a clear limit: when the direct path between start and goal is reached, no further efficiency improvements can be made. The current findings outline a minimal set of cognitive abilities that is necessary and sufficient for cumulative route improvements to emerge.

The identified minimal set of cognitive abilities predicts that species with similar architectures could also show cumulative improvements, with a potential example in social ants who navigate along idiosyncratic one-way routes using landmarks [35]. My results also demonstrate a role for naive individuals in cumulative improvements. This aligns with findings from bluehead wrasse, which use traditional mating sites (like paired agents stick to idiosyncratic routes), but adopt new sites upon complete population replacement [36,37]. It also aligns with empirical work in great tits, in which population turnover drove cumulative improvements in efficiency due to new naive individuals adopting efficient variants [38].

### Do the current findings extend to cumulative cultural evolution?

Examples of behaviour described as "cumulative culture" in animals often do not meet core criteria of a particular framework [19], although cumulative route improvement in pigeons has often been interpreted as meeting all core criteria [16,19] at least for Type I cumulative culture [4]. While the current model reproduces cumulative route improvements in successive generations, it could be argued that its "innovation" is unlike human innovation, and that its "social learning" is without the traditional demonstrator or observer.

Not all human cumulative culture is Type II. For example, humans can increase a wheel's speed over several generations without gaining understanding of the physics behind their solutions [39]. Another example is language, in which systematic structure can develop over generations that share the goal of communicating effectively for the benefit of naive individuals [40]. In these situations, there are clear goals (meaning efficiency can be optimised), memory in experienced agents, and transfer to naive agents through implicit means. While the current model does not readily extend to these situations as it is, there is opportunity to explore the overlap between following another individual along a route and following an individual's actions or utterances.

The difference between cumulative culture in humans and other animals is typically described as a qualitative distinction [4,19]. Some simulations suggest that this distinction could arise from a quantitative difference in the fidelity of information communication [41]. Specifically, high fidelity reduced the loss rate of cultural traits, and if this breached a threshold

it allowed traits to survive long enough to be recombined, which in turn led to cumulative culture. Through an optimistic lens, this could be taken to suggest that rudimentary aspects of cumulative culture found in animals are on the same continuum as cumulative culture found in humans. However, even if that holds for other nonhuman individuals, the artificial navigators introduced here fall well below the required fidelity threshold. Hence, if someone did consider the current model an example of rudimentary "cumulative culture," it would never pass beyond simple optimisation of efficiency.

## Conclusions

In sum, artificial agents with minimal cognition reproduce cumulative route improvements previously shown in pigeons. This is qualitatively different from cumulative culture in humans, and it is unclear whether the current model extends to more complex situations. However, these findings do suggest that cumulative improvement across generations could be an emergent property in animals that work towards a goal alongside more experienced individuals.

## Materials and methods

### Artificial navigator model

Artificial navigators were agents that embarked on journeys from a set starting point to a set goal, although they did not always reach this goal. They were bound by 4 rules, each implemented as an iterative sampling process from a Von Mises distribution. The centre of each distribution was determined by a bearing, and the spread by certainty of information. At each time point, an agent's heading was updated by sampling each distribution and computing a weighted circular mean (Eq 1). Weights were set at agent initialisation and added up to 1. Precision parameters were based on empirical data (see under "Experimental design").

The first rule was **goal direction**. The centre of this distribution was the bearing towards the goal $b_{goal}$, its precision parameter was $\kappa_{goal}$, and its weight $w_{goal}$. The bearing was computed from the coordinates of the goal $(x_{goal}, y_{goal})$ and agent at time $t$ $(x_t, y_t)$ (Eq 2). The purpose of this rule was to orient agents towards the goal, just like pigeons can orient homewards upon being released from unfamiliar sites. This ability likely depends on the sun, as starlings and pigeons can learn to use light and time-of-day to orient towards rewards [21], and pigeons orient homeward when the sun is visible [23]. They can even do so when it is overcast, but their initial orientation becomes more random when magnets are glued to the back of their heads [23], suggesting that pigeons also use an internal compass. For more comprehensive overviews, see [22] and [29].

$$b_{goal} = atan_2(y_{goal} - y_t, x_{goal} - x_t) \tag{2}$$

The second rule was **social proximity**. This distribution is a weighted composite of a Von Mises distribution for social convergence that is centred on the bearing towards another agent's estimated future position $\hat{b}_{other}$ and another Von Mises distribution for social alignment that is centred on another agent's current relative heading. The alignment of headings between agents at close proximity is a crucial part of flocking behaviour [42], but at larger distances agents need to converge rather than align to achieve social proximity. Samples drawn from the convergence distribution were weighted with proportion $p$ and those drawn from the alignment distribution with $(1-p)$. Proportion $p$ was drawn from a cumulative normal distribution with mean 0.5 and standard deviation 0.1, which is equivalent to a distance of 30 metres, at which pigeons are estimated to be able to recognise individuals [43]. Both composite distributions have precision parameter $\kappa_{social}$, and the combined distribution has weight $w_{social}$.

Bearings towards other agents were computed from an agent's position at time $t$, $(x_t, y_t)$ and other agent $j$'s expected position at time $t+1$ (Eq 3). The expected position of agent $j$ at time $t+1$ was estimated on the basis of their velocity $v$ (which was kept constant) and their heading $h_{j,t}$ at time $t$ (Eq 4).

$$\hat{b}_{other} = atan_2(\hat{y}_{j,t+1} - y_t, \hat{x}_{j,t+1} - x_t) \tag{3}$$

$$(\hat{x}_{j,t+1}, \hat{y}_{j,t+1}) = (x_{j,t} + vcos(h_{j,t}), y_{j,t} + vsin(h_{j,t})) \tag{4}$$

The third rule was **route memory**. This was established during an agent's first journey, in which passed landmarks were committed to memory. Across the map of 200 by 130 units, 6,500 landmarks were spread. This aligns with landmark detection using pigeon flight routes [26] and edge detection in aerial photography [25]. During consecutive journeys, an agent attempted to fly from one memorised landmark to the next by sampling from a Von Mises distribution centred on the bearing towards the next landmark $b_{landmark}$, with spread $\kappa_{memory,i}$ for journey $i$, and weight $w_{memory}$ (Eq 5; see Fig C in S1 File). There were no memorised landmarks in the first journey, so the spread for $\kappa_{memory,1}$ was set to 0, resulting in a completely uniform distribution. For all following journeys, $\kappa_{memory,i}$ was set to 1.82, 2.29, 2.98, 4.19, and then plateaued at 6.78. This was analogous to a linear decrease in standard deviation from 0.9 to 0.4 and was based on model fits to pigeon homing data (see under "Data reduction and statistics: Pigeons"). Agents proceeded to navigate towards the next landmark $l+1$ if they came within a threshold distance of landmark $l$. This threshold was set as 10 times the distance agents could travel between time $t$ and time $t+1$.

The gradual improvement in memory precision over several journeys and the anchoring to landmarks were based on Gaussian process models of pigeon navigation [26]. While the current implementation was less elegant than its inspiration, it was computationally inexpensive, and parsimonious with sampling from distributions of other bearings.

$$b_{memory} = atan_2(y_{landmark,l} - y_t, x_{landmark,l} - x_t) \tag{5}$$

The fourth and final rule was **continuity**. This assured that during journey $i$, an agent's next heading at time $t+1$ would be similar to their heading at time $t$. The continuity component was sampled from a Von Mises distribution centred on current heading $h(t)$, with precision parameter $\kappa_{continuity}$, and weight $w_{continuity}$.

Finally, agents set their next heading by drawing random samples $a$ from each of the Von Mises distributions described above and computing their weighted circular mean (Eqs 6–8).

$$h(t + 1) = arctan_2(\bar{y}, \bar{x}) \tag{6}$$

Where:

$$\bar{y} = sin(a_{goal})w_{goal} + sin(a_{other})w_{social} + sin(a_{memory})w_{memory} + sin(a_{continuity})w_{continuity} \tag{7}$$

$$\bar{x} = cos(a_{goal})w_{goal} + cos(a_{other})w_{social} + cos(a_{memory})w_{memory} + cos(a_{continuity})w_{continuity} \tag{8}$$

Software was implemented in Python (version 3.10.12) [44] (for tutorials, see [45,46]), using external libraries Matplotlib (version 3.8.2) [47], NumPy (version 1.26.3) [48], SciPy (version 1.12.0) [49], and utm (version 0.7.0) [50].

## Experimental design

Agents travelled in 3 conditions that mapped onto work in pigeons [16]: solo, paired, and in an experimental condition with generational turnover. In the solo and pair conditions, 1 or 2 agents made 60 consecutive journeys. In the experimental condition, a naive replaced an experienced agent every 12 journeys.

Agents travelled 1 distance unit per 1 time unit, attempting to find a fixed goal from a fixed starting point that were 104 units apart. The maximum distance agents were allowed to travel was 2,506 units. Compared to the map used by pigeons in Sasaki and Biro's study [16], this is equivalent to a flight of 200 km and approximately 5 h. This cut-off was chosen because pigeons would have suffered continuously increasing concentrations of uric acid and other metabolites [51], and a marked increase in reactive oxygen metabolites and decrease in serum antioxidant capacity [52].

Weight parameters $w_{goal}$ and $w_{social}$ varied from 0.05 to 0.35 in steps of 0.05, and $w_{memory}$ from 0.05 to 0.5 in steps of 0.05, resulting in 610 unique combinations. No combinations with weight sums over 1 were included, and $w_{continuity}$ made up the difference for all weight sums under 1. A total of 50 repetitions were done for each condition and each unique combination of parameters, resulting in a total of 30,500 simulations.

Precision parameters were fixed at $\kappa_{goal} = 1.54$ (equivalent SD = 1.0), $\kappa_{social} = 2.18$ (0.80), $\kappa_{memory,1} = 1.82$ (0.9) to $\kappa_{memory,5} = 6.78$ (0.40), and $\kappa_{continuity} = 8.69$ (0.35); roughly based on model fits for stable pigeon pairs (see under "Data reduction and statistics: Pigeons"). This data [53] was published alongside an analysis on leadership in pairs of naive and experienced pigeons [20], and seems to have been the source data for an earlier publication on generational improvements in efficiency [16].

## Data reduction and statistics: Pigeons

Individual pigeon GPS data (defined by latitude and longitude) published by others [53] was converted to Universal Transverse Mercator (UTM) coordinates (grid zone 30U). Samples with velocities under 25 or over 150 km/h were excluded from flights, to filter breaks and apparent GPS glitches. Flights were completely excluded it they contained coordinates further than 17.03 km (twice the start-goal distance) away from the point midway between start and goal. Out of 2,176 files in the original dataset, 6 were excluded for straying too far off course, and 45 for not reaching the goal. Sasaki and Biro [16] also imputed several early incomplete flights with direct-to-home trajectories, which was not done here to avoid fitting models to imputed data, but the pattern of results matches nevertheless (Fig 2).

Best parameter fits for pigeon flight data were determined through maximum likelihood estimation. This is an established way of deriving parameter estimates for mixture models of Von Mises distributions, for example, in research on visual short-term memory [54]. To speed up the fitting process, GPS data was downsampled to 0.05 Hz (1 sample every 20 s).

Averages (standard errors and ranges) for weight parameter estimates in pigeons were $w_{goal}$ = 0.12 (0.03, [0.04–0.34]), $w_{social}$ = 0.16 (0.03, [0.04–0.28]), $w_{memory}$ = 0.09 (0.01, [0.01–0.17]), and $w_{continuity}$ = 0.59 (0.03, [0.49–0.74]) in the experimental condition ($N$ = 10 repetitions of 5 generations each); $w_{goal}$ = 0.04 (0.01, [0.00–0.09]), $w_{social}$ = 0.32 (0.10, [0.13–0.65]), $w_{memory}$ = 0.16 (0.04, [0.04–0.33]), and $w_{continuity}$ = 0.47 (0.11, [0.13–0.75]) in the pair condition ($N$ = 6 repetitions of 60 flights with 2 birds each); and $w_{goal}$ = 0.31 (0.04, [0.15–0.58]), $w_{memory}$ = 0.27 (0.06, [0.02–0.55], and $w_{continuity}$ = 0.43 (0.06, [0.25–0.83]) in the solo condition ($N$ = 9 repetitions of 60 flights with 1 bird each).

Averages (standard errors and ranges) for spread parameter estimates in pigeons were $SD_{goal}$ = 1.16 (0.08, [0.86–1.67]), $SD_{social}$ = 1.04 (0.21, [0.18–1.87]), $Sd_{memory,1}$ = 1.50 (0.21,

[0.52–2.37]) to $SD_{memory,5}$ = 0.26 (0.08, [0.10–0.85]), and $SD_{continuity}$ = 0.33 (0.03, [0.21–0.54]) in the experimental condition; $SD_{goal}$ = 0.85 (0.10, [0.56–1.20]), $SD_{social}$ = 1.03 (0.38, [0.21–2.61]), $SD_{memory,1}$ = 0.73 (0.26, [0.10–1.68]) to $SD_{memory,5}$ = 0.30 (0.11, [0.10–0.77]), and $SD_{continuity}$ = 0.44 (0.04, [0.32–0.64]) in the pair condition; and $SD_{goal}$ = 0.69 (0.14, [0.10–1.37]), $SD_{memory,1}$ = 0.85 (0.15, [0.46–1.69] to $SD_{memory,5}$ = 0.75 (0.24, [0.16–2.02]), and $SD_{continuity}$ = 0.33 (0.08, [0.10–0.78]) in the solo condition. Note that these were fitted as precision ($\kappa$) parameters, but due to their nonlinear scale, I opted to report standard deviation equivalents for clarity.

### Data reduction and statistics: Agents

Simulation results were averaged between paired agents and over independent runs within the same condition and parameter settings. Efficiency for the final generation was computed as the highest out of 12 journeys in the fifth generation.

Generational efficiency improvement was computed as the average difference in route efficiency between consecutive generations. To reduce the impact of random fluctuations, the most efficient (typically the final) routes were taken as representative within each generation. The first generation in the experimental condition was omitted, to avoid comparisons between single and paired journeys.

To avoid trivial statistical significance that can be achieved through increasing the number of simulations, inferences on the basis of statistical tests were avoided and were instead made on the basis of holistic interpretation. Readers are invited to scrutinise figures, data, models, and software.

## Supporting information

**S1 File. PDF document containing Figs A–D.** A single PDF file is provided with figures that offer additional information. Fig A shows measures of efficiency as a function of the $w_{goal}$, $w_{social}$, and $w_{memory}$ components. Fig B shows the relationships between component weights and efficiency outcomes. Fig C shows example journeys and the landmarks that individuals used across generations. Fig D shows histograms of final efficiency and the intergenerational changes in efficiency for each condition.
(PDF)

## Acknowledgments

Thanks to Dr. Paul E. Smaldino, Dr. Takao Sasaki, and members of the Cultural Evolution Discord for feedback on an earlier version of this manuscript.

## Author Contributions

**Conceptualization:** Edwin S. Dalmaijer.

**Data curation:** Edwin S. Dalmaijer.

**Formal analysis:** Edwin S. Dalmaijer.

**Investigation:** Edwin S. Dalmaijer.

**Methodology:** Edwin S. Dalmaijer.

**Software:** Edwin S. Dalmaijer.

**Validation:** Edwin S. Dalmaijer.

**Visualization:** Edwin S. Dalmaijer.

**Writing – original draft:** Edwin S. Dalmaijer.

**Writing – review & editing:** Edwin S. Dalmaijer.

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
