## [Editor Report · Decision Letter 0]

4 Aug 2023

Dear Dr Dalmaijer, 

Thank you for submitting your manuscript entitled "Cumulative culture spontaneously emerges in social navigators with imprecise memory" for consideration as a Short Report by PLOS Biology.

Your manuscript has now been evaluated by the PLOS Biology editorial staff, as well as by an academic editor with relevant expertise, and I'm writing to let you know that we would like to send your submission out for external peer review.

Once your full submission is complete, your paper will undergo a series of checks in preparation for peer review. After your manuscript has passed the checks it will be sent out for review. To provide the metadata for your submission, please Login to Editorial Manager (https://www.editorialmanager.com/pbiology) within two working days, i.e. by Aug 08 2023 11:59PM.

Kind regards,

Roli Roberts

Roland Roberts, PhD

Senior Editor

PLOS Biology

rroberts@plos.org

---

## [Decision Letter · Decision Letter 1]

27 Sep 2023

Dear Dr Dalmaijer,

Thank you for your patience while your manuscript "Cumulative culture spontaneously emerges in social navigators with imprecise memory" was peer-reviewed at PLOS Biology. It has now been evaluated by the PLOS Biology editors, an Academic Editor with relevant expertise, and by four independent reviewers. 

You’ll see that reviewer #1 is positive but wants more a substantial Intro and Discussion, some sensitivity analyses, and has some requests for clarification. Perhaps more seriously, s/he also thinks that your claim that goal direction is unnecessary cannot be true (and even did some modelling to explore this). Reviewer #2 is also positive, but like rev #1 wants significantly more meat in the Intro and Discussion; s/he also thinks that you are too uncritical about the claims of cumulative culture by Sasaki & Biro, and need to justify your choice of definition more rigorously. Reviewer #3 also wants broader context in the Intro, some exploration of parameter space where predictions fail to match empirical data, wonders what happens if only the first generation knows the goal location, and wants more clarity about how heading h is calculated. Like the other reviewers, reviewer #4 also has problems with how the literature is covered (especially human cumulative culture), makes a similar point to rev #2 about the lively debate in the field about definitions, and asks what the “continuity” parameter would correspond to in tasks other than navigation (and whether it’s essential for the findings to hold).

We now invite our reviewers to cross-comment on each other's reviews; during this process, at least one other agreed with the main concern raised by reviewer #1. Several re-emphasised the need for a more substantial Intro and Discussion, and the Academic Editor and I think that your article should be expanded to a full Research Article to give you the space to expand on the issues identified. Also, in case it's helpful, one reviewer said "I completely forgot to add this to the review, but there is a special issue in Phil Trans on CCE across species that would be a good starting place for the author to check https://royalsocietypublishing.org/toc/rstb/2022/377/1843"

In light of the reviews, which you will find at the end of this email, we would like to invite you to revise the work to thoroughly address the reviewers' reports.

Given the extent of revision needed, we cannot make a decision about publication until we have seen the revised manuscript and your response to the reviewers' comments. Your revised manuscript is likely to be sent for further evaluation by all or a subset of the reviewers.

**IMPORTANT - SUBMITTING YOUR REVISION**

*Re-submission Checklist*

*Published Peer Review*

*PLOS Data Policy*

*Blot and Gel Data Policy*

Sincerely,

Roli Foberts

Roland Roberts, PhD

Senior Editor

PLOS Biology

rroberts@plos.org

REVIEWERS' COMMENTS:

Reviewer #1:

The author presents a mathematical simulation of the cumulative cultural improvement of route efficiency, as documented empirically in pigeons. They find that the sort of cumulative improvement observed empirically can be produced under specific parameter values. I very much enjoyed reading this paper, and I would like to see it published eventually. However I have a number of quite significant concerns that I will detail below, and so I will request that significant revisions are made.

Major points:

1. The introduction is far too short. I'm all for getting straight to the point, but the current manuscript does so in such a hurry that it skips over a lot of context that would help get across why the reader should be interested in this work. I'm not going to tell the author how to write an introduction (I assume they are well versed in this), but as one example, it seems pertinent to cover a few other theories about what cognition supports cumulative culture and why those might not be the whole picture. The introduction should be at least double its current length, and could comfortably be much longer still.

2. Introduction paragraph 6. The spread (i.e. precision) parameters are fixed according to an analysis, but there is no reporting of the uncertainty in these estimates. Measures of such uncertainty should be provided. Even better would be a sensitivity analysis - does cumulative culture depend on the specific values used?

3. Results regarding lesions of the goal directed component: "If they found the goal, this resulted in low-efficiency paths with ample room for inter-generational improvement. Indeed, inter-generational efficiency improvement still occurred in the experimental condition, but not in pair or solo controls. Hence, goal direction was not necessary for cumulative culture."

I found this result to be quite surprising, so surprising I think it must be incorrect. I've even built a few simple models to reproduce it while reviewing the manuscript and cannot do so. The issue is that that without the goal directed component I cannot see how there can be any means by which efficiency can ratchet upwards. For instance, suppose one pigeon completes a random walk, and they are then paired with another. The new pigeon will learn from the experienced pigeon, and will superimpose its own random walk on top, but this should be just as likely to worsen the route as to improve it. At first, I wondered if the information was coming from failed routes (those that never reach the goal) being discarded, but the manuscript makes clear this is not the case. Another option is pigeons are initialized facing in the right direction on their first run (making the random walks not truly random). Or perhaps, this is an artefact of the efficiency measure: the initial random walks are at such low efficiency that the only way is up. But maybe I am wrong - if so the author needs to really dig into what is going on here and where the information is coming from that enables the population of achieve progressively higher efficiency.

4. Discussion. Like the introduction the discussion is really short. There needs to be more of an effort to expand on how this work contributes to and interacts with other parts of the literature.

Minor points:

1. Introduction paragraph 3. There are semicolons between the first three rules, but a period (full stop) between the 3rd and 4th. Any reason for this inconsistency?

2. Introduction paragraph 3. Continuity is introduced but not defined, leaving the reader to ponder what it means.

3. Introduction paragraph 4. Von Mises distributions are introduced but not explained. I had to look them up on Wikipedia. It would be perfectly easy to explain they are basically circular normal distributions. It is important to explain this because these distributions are a core part of the model.

4. Introduction paragraph 4. The existence of b_goal needs more justification. As it is, it is magic knowledge of where the goal is. The author needs to justify why it is reasonable to simulate agents with this kind of knowledge. A more detailed discussion of solar/magnetic compasses would be a good step.

5. Introduction paragraph 4, and throughout. The k parameter is defined as a spread, but it's not, it's the reciprocal of a spread, more intuitively called concentration or precision.

6. Introduction paragraph 4. The author states that uncertainty in the location of the goal is greater than uncertainty in the location of the next landmark. Is this an assumption of the model or an empirically documented fact? It needs greater explication and justification.

7. Introduction paragraph 5, and throughout. It is never really stated that the experimental condition also involves pairs. Rather, turnover is mentioned, but not the group size.

8. Introduction paragraph 6. The term "runs" needs to be more clearly defined.

9. Methods. Is it plausible that b_other pulls you towards the other pigeons projected location? At high enough weights this will cause pigeon collisions. My recollection of the flocking literature is that individuals align their headings with those of nearby individuals, and not that they explicitly fly at them.

10. Methods. Route memory. It is important to make clear here that the landmarks are therefore unique to each agent. This is not intuitive, as in the real-world landmarks are typically conspicuous features (a lake, or tower etc) and so agents cannot freely choose their locations. But here, part of the reason that efficiency can improve is that each agent generates its own landmarks from a continuous landscape allowing for gradual improvements. This sort of thing would be suitable for an expanded discussion.

11. Methods. Experimental design. It is not clear how the choice of speed (70 units per 1 time unit) affects the results. This needs an expanded justification (or clarification of why it doesn't matter). If it does matter, a sensitivity check would be helpful.

12. Methods. Experimental design, Weight parameters. How are all these weight parameters possible? Many combinations would exceed a sum of 1? In the supplementary figures it looks like only valid combinations are considered, but this needs to be stated.

13. Methods, Data reduction. How many samples (and what %) were discarded for either GPS glitches or excessive distance from the route?

14. Methods, Data reduction. There is a reference to "the original paper", but which paper is this? What exactly is the subtle deviation? How can we be sure it doesn't matter?

15. Methods, Data reduction. Why is efficiency computed from the best of 12 journeys? Why not the average across all 12? Does this matter?

16. Figures S1 and S2. These figures are pretty, but the nested circles render them almost incomprehensible. I'm not sure what the best solution is, but perhaps separating the plots by w_memory is the only option.

Reviewer #2:

This article, "Cumulative culture spontaneously emerges in social navigators with imprecise memory", describes an avian navigation model where agents follow just four rules. Navigators that were paired outperformed solo navigators, and when the composition of the pairs was varied rather than fixed, efficiency improved even further. Interestingly, both the experienced and naive individuals benefitted from each other, which was somewhat unexpected: one would not think (at least initially) that the naive individual would be of much benefit to the experienced one. 

This model and the findings of this paper pose a provocative question to the field of non-human culture, where there is a lot of debate currently about this topic (see Whiten 2023 Physics of Life Reviews, and the slew of responses). I think there is a lot of value in papers like this, which challenge the prevailing lines of thought in the field, which has at times become stagnated, and has always been too anthropocentric for its own good. It's an interesting time for the field, with a lot of new findings emerging (or on the brink of doing so). So, I would be happy to see this article published, and I do think that PLOS Biology would be a good fit. However, I do have some comments for the authors to consider first.

I am admittedly not an expert in modelling, so cannot really comment on the exact ins and outs of the model itself - although I can't see any glaring issues with it, the parameters are quite thorough and the controls seem reasonable. Hopefully, the other reviewers will be able to comment more thoroughly on this. My comments will therefore focus mainly on the introduction and discussion, which are at present, very brief, and could use expansion (or, at least reference to certain points - I am aware this is a short report, so won't necessarily be able to include much depth, but at least referencing would be helpful). 

The original paper by Sasaki and Biro, which involved releasing pigeons (as singles or pairs, of either fixed or variable composition) is one of my favourites. While I am personally inclined to agree that Sasaki and Biro demonstrat

---

## [Decision Letter · Decision Letter 2]

10 Apr 2024

Dear Dr Dalmaijer,

Thank you for your patience while we considered your revised manuscript "Cumulative culture spontaneously emerges in artificial navigators that have goal-direction and route memory, and seek social proximity" for publication as a Research Article at PLOS Biology. This revised version of your manuscript has been evaluated by the PLOS Biology editors, the Academic Editor and the original reviewers.

Based on the reviews, we are likely to accept this manuscript for publication, provided you satisfactorily address the remaining points raised by the reviewers and the Academic Editor. Please also make sure to address the following data and other policy-related requests.

IMPORTANT - Please attend to the following:

a) Please address all of the remaining requests from the reviewers.

b) The Academic Editor has also provided some comments, in which s/he firmly requests that you avoid the term "cumulative culture" for most of the paper, including the Title.

c) After some discussion with my team-members, who found the current Title rather obscure, and with the Academic Editor, we suggest that you change it to something like "Cumulative route improvements spontaneously emerge in artificial navigators even in the absence of sophisticated communication or thought" (perhaps "in silico navigators" would be clearer?)

d) My team-mates also found the Abstract rather difficult to comprehend, and this will need to be improved, especially for our broader readership. Specifically, two things were very unclear. Firstly, what are the "artificial agents" that you use - my understanding is that these are in silico or simulated entities, and this should be made clear in language that will be clear not only to those in your own field but also to general biologists (the majority of our readership). The second thing, which does not seem to appear in your abstract, is that you compare your model to real-life data from pigeons, finding a good fit - this should be mentioned in the Abstract, and overall the Abstract should be re-worked with the broader reader in mind.

e) you say in the financial declaration that you received no specific funding for this work. Can I just confirm that this is the case?

f) Many thanks for supplying the data and code. My understanding (correct me if I'm wrong) is that the data are in Zenodo, and that the code is in Github. Because Github depositions can be readily changed or deleted, please make a permanent DOI’d copy of the code (e.g. in Zenodo) and provide this URL (see below).

g) Please cite the location of the data clearly in all relevant main and supplementary Figure legends, e.g. “The data underlying this Figure can be found in https://zenodo.org/records/XXXXXXXX" or "The data and code required to generate this Figure can be found in https://zenodo.org/records/XXXXXXXX"

We expect to receive your revised manuscript within two weeks. 

*Published Peer Review History*

*Press*

Sincerely,

Roli

Roland Roberts, PhD

Senior Editor

rroberts@plos.org

PLOS Biology

REVIEWERS' COMMENTS:

Reviewer #1:

The author has revised and resubmitted their model of the cumulative cultural evolution of efficient flight paths in pigeons. I previously enjoyed reading this paper and continued to do so following its revisions. The author should be commended for their thorough improvements to the manuscript. I was particularly appreciative of their deep dive into the role of goal-direction and the results of the model now make sense (to me at least). Below I have a few minor concerns, however, overall I have no reservations in recommending this manuscript for publication.

L.54. Through the symbol for b^_other does not render correctly, appearing as an empty cube.

L55. Typo. "in components' precision" should be "in each component's precision" (note shift of apostrophe).

E1. It is not clear why k_memory is indexed by i. This is explained in the methods, but I would mention here it is because the precision of memory increases across flights.

L90-98. Typos. W_goal and w_social are presented as integers (e.g. 15) instead of decimals (e.g. 0.15).

L129-150. This section is hard to follow and could be improved. Specifically:

L130-132. What is meant by "identical" is it that the same weights are used?

L135. Typo. ", .".

L136-137. Note that this necessity can be seen in that efficiency did not improve when the goal-direction system was lesioned.

L157. In what way do these results not align with agents?

L201. "demonstrates" should be "demonstrate".

Reviewer #2:

I would like to thank the author for their clear and detailed responses, particularly to the more technical modelling queries. For my part, the rewritten article has addressed my concerns regarding the lack of background information and need for wider context, and I am happy to recommend it be published in its current form. Although, as I mentioned previously, I'm no expert in modelling, so please do defer to the more knowledgeable reviewers' opinions regarding the responses to these queries. However, I did find the author's responses to these questions (and changes made as a result) to be clear and plausible to me, and I at least was satisfied that they were addressed sufficiently.

Reviewer #3:

Many thanks to the authors for their clear responses to my comments--the manuscript is greatly improved in clarity and contribution, and I can recommend this for publication. I especially liked that the ms now highlights the mechanism as regression to the goal, and is a nice formalization of this mechanism that can lead to CCE.

I only have some lingering minor comments below, which I do not need responses to:

L133: "instead of goal-direction, social proximity, route memory, or continuity Von Mises distributions, headings were sampled from uniform distributions" 

"continuity Von Mises distributions" sounds wrong here and might confuse readers. Suggest instead to avoid it: However, headings were sampled from uniform distributions (i.e. noise) instead of being influenced by goal-direction, social proximity, route memory, or continuity.

L179: "While focussing on task efficiency offers insight into cumulative cultural evolution (Gruber et al., 2021), solutions are only one side"

I think authors meant "one-sided", although I still don't quite get what that means. Maybe meant to say something like "evolutionary paths are limited?"

L203: "but adopt new sites upon population refresh"

I like the turn of phrase, but maybe something like "upon complete population replacement" would be clearer.

L205: "naive individuals conforming to efficient variants"

Would recommend "adopting efficient variants", since as I remember there was no evidence of conformity in this experiment

L330-345: all of these reported metrics might be better off in a table, and in the supplementary. Will save you some space.

Reviewer #4:

The paper has shown clear improvements, and I believe the author has appropriately addressed most of the raised comments. However, there are still some areas that need attention:

Major comments:

While the revised introduction does a better job of citing existing work, the structure remains poor, making it difficult to follow. For example, the first sentence defines cumulative culture, followed by a sentence about underlying mechanisms (line 6), then a discussion on cumulative culture in animals (lines 8-12), followed by paragraphs on alternative mechanisms/explanations (lines 13-17), and more definitions of cumulative culture later on. I recommend the author streamline the material from line 1 to 36 for greater clarity.

The author cites the Mesoudi & Thornton definition of cumulative culture to argue that the documented phenomenon constitutes an instance of cumulative culture. However, the second criterion of this definition specifies that the behavioral variant must be passed to others via social learning. Typically, this involves a cultural demonstrator (an individual with experience) and a cultural learner (an individual naive to the task). Here, the only social mechanism is social proximity. I am not convinced this qualifies as social learning, which should be discussed further. As I mentioned in my previous review, it is crucial to articulate these results within the existing literature.

Minor comment:

Line 135: Part of the sentence/paragraph is missing.

COMMENTS FROM THE ACADEMIC EDITOR:

I do find, however, that the author has not sufficiently addressed reviewer 2's comment - that the paper is not in fact about cumulative culture by most people's reading of the term. Readers might be misguided by the title to think that a human-like phenomenon is under examination: the building of new technological or cultural innovations on previously existing ones. Instead it refers to a single study on gradual flight route improvements in pigeons that come about by social learning. The authors of that study have indeed argued that this qualifies as cumulative culture, but this is not widely accepted: indeed it does not contain the innovation of any novel behavioural routines (e.g. the use of a tool that spreads via social learning) - much less the building of a further unique innovation on top of another. There are more impressive examples about the gradual improvement of spatial information in animal groups - e.g. the consensus decision making in honeybee swarms (cf work by Martin Lindauer in Tom Seeley), and no one has seen the need to label these as cumulative culture. I don't think that the modelling in the present study would generalise to other cases than the particular phenomenon in pigeon navigation. So I think the author would do himself a favour if he removed "cumulative culture" from the title and instead indicated a clear focus on the specific phenomenon under investigation: route improvements in navigation via social learning. The same applies to elsewhere in the study. It's fine to discuss the study in its relation to cumulative culture, but readers will be disappointed if it's pitched as it currently is.

[...and in response to Roli Roberts' suggestion that the Title should be changed to something like "Cumulative route improvements spontaneously emerge in artificial navigators even in the absence of sophisticated communication or thought" and the whether it would be fair/appropriate to ask the author to substitute "cumulative route improvements" for "cumulative culture" throughout...:]

....yes, you have my full support for this change of the title, and the terminology throughout. I think an interesting finding has to stand on its own no matter whether one sticks a fancy label on it or not. I think it's fine if he wants to have a brief discussion of cumulative culture somewhere, e.g. "such cumulative route improvements have even been discussed as bearing some of the elements of cumulative culture as occurs in species X, Y and Z; I suggest that my modelling might also apply to phenomenon x if the following conditions are met" or something like this.

---

## [Editor Report · Decision Letter 3]

26 Apr 2024

Dear Dr Dalmaijer,

Thank you for the submission of your revised Research Article "Cumulative route improvements spontaneously emerge in artificial navigators even in the absence of sophisticated communication or thought" for publication in PLOS Biology. On behalf of my colleagues and the Academic Editor, Lars Chittka, I'm pleased to say that we can in principle accept your manuscript for publication, provided you address any remaining formatting and reporting issues. These will be detailed in an email you should receive within 2-3 business days from our colleagues in the journal operations team; no action is required from you until then. Please note that we will not be able to formally accept your manuscript and schedule it for publication until you have completed any requested changes.

Sincerely, 

Roli Roberts

Senior Editor

PLOS Biology

rroberts@plos.org